# BANDIT-MOE:
# DIVERSE KNOWLEDGE ACQUISITION THROUGH BANDIT ROUTING FOR CONTINUAL LEARNING

## ABSTRACT

Substantial updates to the parameters of deep learning models constitute a prominent factor underlying catastrophic forgetting in the continual learning. To tackle this challenge, the Mixture-of-Experts (MoE) framework has been introduced into continual learning to leverage its routing strategy to select parts of relevant experts for training, thereby mitigating parameter overwriting. However, in continual learning, the routing strategy tends to allocate tasks to a small number of highly optimized experts trained on prior samples, which results in the overwriting of favored experts while rendering other experts underutilized. Therefore, we formulate expert routing in MoE as a Multi-Armed Bandit problem and propose the Bandit-MoE framework. It consists of a Bandit Routing (BR) strategy and a specific expert structure. BR estimates the maximum expected gain for each expert by incorporating both the expectation and the variance of the reward for the incoming samples. This strategy significantly reduces the early neglect of certain experts and ensures a more balanced expert selection, thereby improving knowledge preservation. Finally, a comprehensive series of experiments are conducted to investigate the impact of expert structures on continual learning. The results of three widely used benchmark datasets have shown that Bandit-MoE consistently outperforms the prior art in all experimental settings, demonstrating the effectiveness of Bandit-MoE for continual learning.

## 1 INTRODUCTION

In the real world, information typically arrives sequentially, requiring models to be continuously updated with subsequent data. Training with both historical and subsequent data requires large computational resources. In contrast, training solely on subsequent data may lead to substantial parameter updates, thereby resulting in knowledge overwriting and forgetting. Mixture-of-Experts (MoE) can be introduced to mitigate catastrophic forgetting in continual learning, due to its routing strategy Shazeer et al. (2017) and experts structure. For any input sample, only a small subset of experts is activated according to the routing strategy. During continual learning, only the parameters of the relevant experts are updated, while those of other experts remain unchanged, thereby minimizing the risk of catastrophic parameter overwriting.

However, current MoE routing strategies Shazeer et al. (2017) tend to allocate tasks to experts with higher immediate performance, which will introduce bias due to uneven optimization states across experts. Consequently, in continual learning, the MoE exhibits preferential knowledge accumulation towards previously trained experts while underutilizing insufficiently optimized ones, establishing a self-reinforcing feedback that progressively amplifies routing bias through iterative training. As a result, only a few experts are repeatedly selected to handle the task at hand, while others have difficulty contributing effectively until they are fully optimized. As shown in Figure 1 (a), during the training phase, existing MoE routing strategies Yu et al. (2024) consistently favor specific experts (e.g., The ninth expert). This can lead to severe parameter overwriting, which in turn causes significant forgetting. Other experts, despite their lower forgetting rates, remain underutilized, thereby compromising the overall performance of the model.

To address these challenges, we formulate the expert selection problem in MoE as a Multi-Armed Bandit (MAB) problem Lai & Robbins (1985); Li et al. (2010). This aligns with the MoE's goal of maximizing performance by optimally choosing among experts, similar to an MAB agent's aim to maximize cumulative rewards by selecting from arms Angela & Dayan (2005). In the MAB

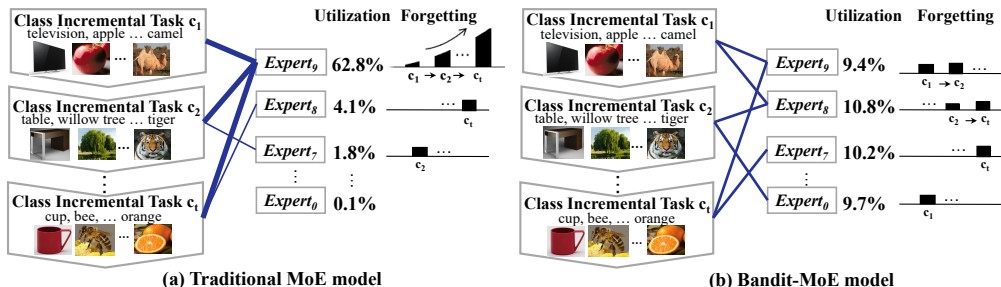

Figure 1: Comparison of different MoE in continual learning. (a) depicts continual learning with a traditional MoE. Since $Expert_9$ is optimized by task $c_1$, it has better current performance than other experts. Subsequent tasks have a preference for $Expert_9$. This leads to frequent parameter overwriting on $Expert_9$ during continual learning, resulting in a high forgetting rate. In contrast, other experts, though not suffering severe forgetting, remain underutilized as they lack sufficient optimization, leaving the MoE under-exploited overall. (b) shows continual learning with the Bandit-MoE. By introducing the BR strategy, data is allocated based on the upper bound of experts' performance, avoiding frequent parameter overwriting on any single expert. Meanwhile, BR has a higher probability of exploiting the most relevant experts, which reduces the magnitude of parameter updates and thus lowers the forgetting rate.

framework, an agent that overly exploits the highest-reward arm might miss out on exploring others with better long-term potential. This insufficient exploration also constitutes a challenge faced by MoE in continual learning. Therefore, we propose the Bandit-MoE framework with Bandit Routing (BR) strategy, which employs the Upper Confidence Bound (UCB) algorithm Auer et al. (2002) to estimate the upper bound of each expert's expected gain, formulating a composite metric with the expectation and the variance of the reward. The expectation reflects the expert's performance, while the variance indicates the unexplored potential of experts. Experts with higher expected gain are more likely to be selected. As shown in Figure 1 (b), the BR strategy promotes knowledge preservation by ensuring a sufficient exploration of the most relevant expert for each task. Furthermore, we theoretically prove that the estimation bias of expected gains remains within bounded limits, thereby ensuring the reliability of the BR strategy. In the ablation study, we evaluate the exploration and exploitation of experts, and compare the forgetting rates of BR and other routing.

In addition to the routing strategy, the ability of each expert network is crucial. Specifically, our goal is to reduce the knowledge overwriting of each expert while maintaining the overall ability of the model. To achieve this, we investigate three types of expert structure: Multilayer Perceptron (MLP) Rumelhart et al. (1986), Low-Rank Adaptation (LoRA) Hu et al. (2022), and Kolmogorov-Arnold Networks (KAN) Liu et al. (2025). For instance, MLP is the simplest and most widely used expert structure. LoRA improves efficiency by updating a low-rank decomposition of the original model weight matrix, thereby reducing both storage and computational costs while preserving the expressiveness of the model. Additionally, we explore the impact of KAN in continual learning. KAN introduces learnable activation functions in neuronal connections, replacing each weight parameter with a spline-based function. The locality of the spline ensures that each sample only influences nearby spline coefficients, thereby improving resistance to forgetting in one-dimensional data Liu et al. (2025). However, its effectiveness on high-dimensional data remains underexplored. Our findings indicate that KAN experts exhibit strong resistance to catastrophic forgetting when the total number of experts is small. However, as the number of experts increases, training KAN experts becomes increasingly challenging due to the complexity of optimizing spline-based transformations. The contributions of this paper are as follows:

- We propose a Bandit-MoE framework based on a BR strategy to efficiently explore and exploit experts. The BR strategy selects experts via upper confidence bounds on their expected gains, preventing premature expert exclusion and ensuring fair selection. Furthermore, we theoretically prove that the estimation bias of expected gains can be bounded, thereby ensuring the reliability of the proposed routing strategy.

- The proposed BR routing is compatible with diverse expert architectures (e.g., MLP, LoRA, and KAN) for continual learning. We have conducted extensive testing and analysis of these three structures to provide insights for future research.

- A series of experiments on the CIFAR-100, ImageNet-100, and TinyImageNet datasets show that Bandit-MoE outperforms prior arts in continual learning, particularly in more challenging long-sequence continual learning.

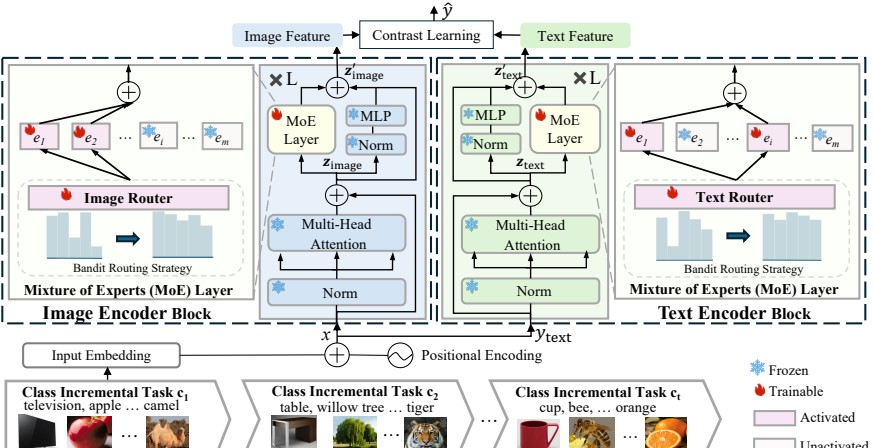

Figure 2: Framework of the Bandit-MoE. The Bandit-MoE is built upon the pre-trained CLIP Radford et al. (2021) and incorporates MoE layers in the MLP position. Each MoE layer consists of one router and $m$ expert networks ($e_1, \ldots, e_m$) to ensure model sparsity. The Bandit Routing (BR) strategy is implemented in the router to promote diverse the knowledge representation within the model. During training, only the router and the selected experts are trainable. L represents the number of blocks in the model. $y_{\text{text}}$ denotes the input to the text encoder, consisting of class name and prompt, such as "a bad photo of a {*class name*}".

## 2 RELATED WORKS

**Continual Learning (CL).** Continual learning methods are generally divided into three categories: regularization-based, replay-based, and architecture-based methods. Specifically, regularization-based methods Li & Hoiem (2017); Ding et al. (2022) introduce explicit regularization terms to balance the learning of new and old tasks. However, these methods often face challenges such as knowledge interference and the risk of converging to local optima. Replay-based methods Aich (2021) utilize rehearsal buffers to store data from previous tasks, which helps mitigate catastrophic forgetting. Nevertheless, their performance tends to degrade as the buffer size decreases, and data storage can pose challenges in privacy-sensitive applications. Architecture-based methods Mallya & Lazebnik (2018); Douillard et al. (2022); Ostapenko et al. (2021) address the CL problem by constructing task-specific parameters and designing specialized model architectures. Unlike other methods that share parameters across tasks and suffer from inter-task interference, architecture-based methods reduce this problem by isolating task-specific knowledge. The MoE framework is a typical architecture-based method. In this paper, we propose a MoE framework for CL that dynamically selects experts through a routing strategy to maximize expected returns. This approach enhances the model's adaptability to new tasks and data during continual learning.

**Mixture-of-Experts (MoE).** The MoE structure consists of two main components: experts and routing strategy Shazeer et al. (2017). In expert learning, studies such as VLMoE Bao et al. (2022), HyperLLaVA Zhang et al. (2024), and MoPE Jiang et al. (2024) employ MLP experts in multimodal large models to enhance performance on downstream tasks. SEED Rypeść et al. (2024) utilizes Gaussian distributions to train each MLP expert to represent different categories, creating heterogeneous experts that help mitigate catastrophic forgetting. However, these methods do not adequately exploit the representation power of the expert networks. In routing strategy, GShard Lepikhin et al. (2021) integrates MoE into the transformer architecture, expanding its practical applications. To reduce computational and communication costs, Switch Transformers Fedus et al. (2022) simplifies the routing strategy by selecting a single expert based on a set capacity factor. In the context of continual learning, MoE dynamically adapts to environmental data and mitigates catastrophic forgetting. However, MoE-Adapters Yu et al. (2024) fails to fully leverage the potential benefits of expert learning in dynamic routing, which will result in overwriting knowledge. In this paper, we propose a novel MoE framework with Bandit Routing (BR) strategy to fully explore and utilize each expert, which is expected to acquire and retain diverse knowledge.

**Multi-Armed Bandit (MAB) Problem.** The Multi-Armed Bandit problem Lai & Robbins (1985); Li et al. (2010) involves selecting from a set of arms, where each pull results in a random reward, and the objective is to maximize the total reward accumulated over a given period. For instance, OWL Kessler et al. (2022) uses reinforcement learning to select independent task heads and formulates the selection strategy as a MAB problem during testing. The primary challenge lies in balancing exploration (trying different arms to gather information about their reward distributions) and exploitation (selecting the arm with the highest expected reward based on current knowledge). Specifically, the problem involves making decisions among competing choices to maximize expected returns, reflecting the dynamic balance between acquiring new information and leveraging existing knowledge Angela & Dayan (2005). In this context, we propose the BR strategy to ensure fair exploration of each expert's potential for the current task, which enhances the efficiency of expert exploitation within the MoE framework. By employing the UCB Auer et al. (2002) algorithm, the BR strategy dynamically evaluates the potential of each expert, ensuring that all experts are comprehensively trained and effectively enhance the model's knowledge retention.

## 3 METHOD

In this section, we describe the Bandit-MoE structure and the BR strategy. We leverage the UCB algorithm from the Multi-Armed Bandit problem to demonstrate the effectiveness of the BR strategy in expert selection.

### 3.1 FRAMEWORK OVERVIEW

Our experiment addresses continual learning for class-incremental tasks, where the task id is not visible during the testing phase. We define each class-incremental task as $C = \{c_1, \ldots, c_t\}$, where the $t$-th task $c_t = \{(x_i, y_i)\}_{i=1}^{I_t}$ consist of input samples $x_i \in X$ and their corresponding labels $y_i \in Y$, with $I_t$ representing the number of samples for the $t$-th task. To address catastrophic forgetting, we employ a MoE model based on the pre-trained CLIP model, as illustrated in Figure 2. Each MoE layer consists of a router and $m$ expert networks. The router, denoted as $R$, has parameters $\boldsymbol{\theta}_R \in R^{l \times m}$, where $l$ is the feature dimension. The outputs of the router are denoted as $[r_1, \ldots, r_m]$, and are defined by:

$$[r_1, \ldots, r_m] \triangleq R(\mathbf{z}_i) = \mathbf{z}_i \cdot \boldsymbol{\theta}_R, \tag{1}$$

where $\mathbf{z}_i \in R^{1 \times l}$ is the $i$-th output of the multi-head attention, and $\mathbf{z} \in \{\mathbf{z}_{\text{image}}, \mathbf{z}_{\text{text}}\}$. Each element of the router output represents the reward for each expert given the input data. The router selects the top $K$ experts based on these rewards, activating only the selected experts while freezing the others to mitigate catastrophic forgetting; that is, $r_i = 0$, except for the top $K$ rewards ($r_i$). The expert networks are denoted as $\{e_1, \ldots, e_m\}$. The output of the experts is formulated as:

$$\mathbf{z}_i' = \sum_{i=1}^{m} r_i e_i(\mathbf{z}_i) + MLP(LN(\mathbf{z}_i)) + \mathbf{z}_i, \tag{2}$$

where $\mathbf{z}_i'$ is the $i$-th output of the encoder block, and $\mathbf{z}' \in \{\mathbf{z}_{\text{image}}', \mathbf{z}_{\text{text}}'\}$. Here, $LN(\cdot)$ denotes layer normalization, and $MLP(\cdot)$ denotes a multi-layer perceptron. Based on $\mathbf{z}_i'$, the predicted value $\hat{y}$ is calculated as follows:

$$\hat{y} = \frac{\exp(\frac{\langle \mathbf{z}_{\text{image},i}', \mathbf{z}_{\text{text},y_i}' \rangle}{\tau})}{\sum_{d=1}^{D} \exp(\frac{\langle \mathbf{z}_{\text{image},i}', \mathbf{z}_{\text{text},y_d}' \rangle}{\tau})}, \tag{3}$$

where $D$ is the total number of categories in the dataset.

### 3.2 THE BANDIT ROUTING STRATEGY

The Multi-Armed Bandit problem Li et al. (2010) involves making decisions among competing choices to maximize expected gains. Similarly, the MoE framework selects experts multiple times to achieve maximum gain. We formulate the routing of MoE as the Multi-Armed Bandit problem. However, some experts may be inadequately trained, making it challenging to estimate their expected gains accurately. To address this, we employ the UCB Auer et al. (2002) strategy to estimate the maximum expected gain of each expert, which serves as the metric for expert selection. The UCB strategy for each expert consists of two components that account for the expert's importance and uncertainty. This approach prevents the premature exclusion of experts, ensures fair selection, and allows the model to acquire a diverse range of expert knowledge. The UCB of the $i$-th expert $e_i$ is defined as follows:

$$\text{UCB}_i \triangleq \tilde{\mu}_i + \tilde{\delta}_i \geq \mu_i, \tag{4}$$

where $\tilde{\mu}_i, \tilde{\delta}_i$ and $\mu_i$ denote the estimated expectation of gain, the estimated variance of the gain and the actual expectation of gain for expert $e_i$, respectively. We set an upper confidence bound for each

expert to measure its uncertainty and current performance. This ensures that all experts have the opportunity to be explored and adequately trained. To ensure that the bias of UCB is bounded, we define the estimation expectation $\tilde{\mu}_i$ and the estimation variance $\tilde{\delta}_i$ as follows:

$$\widetilde{\mu}_i = \frac{\sum_{j=1}^{n_i} reward_{i,j}}{n_i} = \frac{\sum_{j=1}^{n_i} \mathrm{r}_{i,j}}{n_i}, \widetilde{\delta}_i = \sqrt{\frac{2\ln N}{n_i}}, \tag{5}$$

where $\mathrm{r}_{i,j}$ is the reward that expert $e_i$ at the $j$-th selection, $n_i$ denotes the number of times expert $e_i$ has been selected, and $N$ is the total number of samples that have been trained, $i.e.$ the total number of expert selections. In the following proof, we show that the designed UCB for expert gain can accurately estimate the actual gain of each expert, $i.e.$, that the bias of the UCB is bounded.

*Proof.* Let $\mu$ represent the theoretical gain of each expert, where $\mu \in [0,1]$. Let $n$ denote the number of times the expert has been selected, $\mathrm{r}_j$ be the estimated gain of the expert in the $j$-th selection, and $\widetilde{\mu} = \frac{\sum_{j=1}^{} \mathrm{r}_j}{n}$ be the average estimated gain. Let $\delta$ denote the difference between the theoretical and estimated expert gains. Using Markov's inequality Gagniuc (2017) and assuming that the global variance bound can be approximated by the variance bound of independent experiments, we obtain the probability:

$$P\left[\mu > \widetilde{\mu} + \delta\right] = P\left[\sum_i (\mu - \mathrm{r}_j) > n\delta\right] = P\left[e^{\lambda \sum_i (\mu - \mathrm{r}_j)} > e^{n\lambda\delta}\right]. \tag{6}$$

Using Markov's inequality, $P\left[Z > \beta\right] \leq \frac{E[Z]}{\beta}$, when $Z = e^{\lambda \sum_i (p - X_i)}$, and $\beta = e^{n\lambda\delta}$.

$$P\left[\mu > \widetilde{\mu} + \delta\right] \leq e^{-n\lambda\delta} E\left[e^{\lambda \sum_i (\mu - \mathrm{r}_j)}\right]. \tag{7}$$

For any $a \leq Y \leq b$, Hoeffding's Lemma Hoeffding (1963) provides $E\left(e^{\lambda(E(Y)-Y)}\right) \leq e^{\frac{\lambda^2(b-a)^2}{8}}$. In our case, $a = 0$, $b = 1$, so:

$$E\left[e^{\lambda \sum_i (\mu - \mathrm{r}_j)}\right] e^{-n\lambda\delta} = \prod_i E\left[e^{\lambda(\mu - \mathrm{r}_j)}\right] e^{-n\lambda\delta} \leq \prod_i e^{\frac{\lambda^2}{8}} e^{-n\lambda\delta} = e^{\frac{n\lambda^2}{8} - n\lambda\delta}. \tag{8}$$

To minimize the expression $\frac{n\lambda^2}{8} - n\lambda\delta$, we choose $\lambda = 4\delta$, resulting in: $\min_{\lambda > 0} e^{-n\lambda\delta + \frac{n\lambda^2}{8}} = e^{-2n\delta^2}$.

Applying the exponential transformation and Markov's inequality, we obtain the upper bound:

$$P\left[\mu > \widetilde{\mu} + \delta\right] \leq e^{-2n\delta^2}. \tag{9}$$

Taking the inverse probability, the final inequality is obtained:

$$P[\mu - \widetilde{\mu} \leq \delta] \geq 1 - e^{-2n\delta^2}. \tag{10}$$

To ensure that less important experts are explored more frequently, we set $\delta$ to $\sqrt{\frac{2\ln N}{n}}$. Thus, the minimum probability that $\mu \leq \widetilde{\mu} + \sqrt{\frac{2\ln N}{n}}$ is $1 - \frac{1}{N^4}$. As the experiment progresses, the variance of the estimates decreases, and $\mu$ converges to $\widetilde{\mu}$, meaning that the estimated expert gain approaches the actual gain. According to the 3 $\sigma$ principle Pukelsheim (1994), events with a probability greater than 99.73% are high probability events. In our case, when $N = 5$, this probability is $1 - \frac{1}{N^4} = 0.9984$, meaning that after 5 expert selections, the variance of the expert potential is bounded. $\qed$

Therefore, our algorithm is reasonable. For convenience in experimental operations, we implement the BR algorithm directly on the router, integrating it into the update process of the model routing parameters. According to Eq. 4 and 5, the UCB of actual gain for expert $e_i$ can be estimated as follows:

$$\mu_i \leq \frac{\sum_{j=1}^{n_i} \mathrm{r}_{i,j}}{n_i} + \beta \sqrt{\frac{2\ln N}{n_i}}. \tag{11}$$

The hyperparameter $\beta$ is used to adjust the size of the confidence boundaries, thereby controlling the trade-off between exploration and exploitation in the algorithm. Larger values of $\beta$ increase the exploration component, prompting the algorithm to focus more on exploring under-trained experts. Conversely, smaller values of $\beta$ decrease the exploration component, leading to a greater emphasis on exploiting known routing information. Finally, we update the parameters using the cross-entropy loss function:

$$L = -\frac{1}{I_t} \sum_{i=1}^{I_t} y_i \log\left(\widehat{y}_i\right). \tag{12}$$

For continual learning tasks with the BR algorithm, the training parameter dimensions of the image router remain fixed, while those of the text router expand as tasks progress. Therefore, we use the average expert reward for selection in the image router and the expert reward of the model update for the text router to simplify calculations. The pseudocode of the BR algorithm is provided in the Appendix A.1.

### 3.3 EXPERT NETWORKS FOR CONTINUAL LEARNING

We evaluate three types of expert networks (MLP, LoRA, and KAN) by replacing each expert network $e_i$ in Figure 2 and assessing their performance on continual learning tasks.

**MLP**. The Multi-Layer Perceptron (MLP) is a widely used neural network architecture. It is highly scalable and can be adapted to various tasks by increasing the capacity of the pre-trained model. We implement the MLP expert with a single fully connected layer.

**LoRA**. To enhance training efficiency during the fine-tuning phase, we use LoRA as the expert network. LoRA fine-tunes a subset of the model's weights by introducing low-rank matrices, thereby retaining most of the original model parameters. Let $W_{LN+MLP}$ denote the weight matrix of the layer normalization and MLP layer, and $\Delta W_{LoRA}$ denote the weight matrix introduced by LoRA experts. During fine-tuning, LoRA experts decompose this weight matrix into two smaller low-rank matrices, $A$ and $B$. The expert parameters selected during the inference phase are represented as $W_e$:

$$W_e = W_{LN+MLP} + \sum_{i=1}^{K} \Delta W_{LoRA_i} = W_{LN+MLP} + \sum_{i=1}^{K} A_i \times B_i, \tag{13}$$

where matrix $A$ has dimensions $l \times r$ and matrix $B$ has dimensions $r \times l$. Here, $l$ denotes the input and output dimension of the expert, and $r$ represents the rank of LoRA experts in the pre-trained CLIP model, which is considerably smaller than $l$. The parameter $K$ represents the top $K$ experts. Compared to the MLP expert network configuration, the LoRA expert offers higher parameter efficiency and reduced computational cost during fine-tuning, which is utilized in our main experiment.

**KAN**. Additionally, we investigate the performance of KAN as the expert network to evaluate its effectiveness in continual learning, as claimed by previous work Liu et al. (2025). Unlike the MLP and LoRA experts, KAN experts introduce flexibility by employing a learnable activation function between nodes. Furthermore, KAN experts exhibit local plasticity, enabling each connection point to adapt to local features of the input data without altering the parameters of the entire network. We explore whether this local adaptability is particularly advantageous in mitigating catastrophic forgetting in our experiment. The structure of KAN experts is defined as follows:

$$W_e = \left( \Phi_{L-1} \circ \Phi_{L-2} \circ \cdots \circ \Phi_1 \circ \Phi_0 \right), \tag{14}$$

where $\circ$ denotes the sequential application of activation functions or linear transformations layer by layer, and $\Phi = \{\varphi_{i,j}\}, i = 1, \ldots, l$, and $j = 1, \ldots, l$. More details and operational rules can be found in the paper Liu et al. (2025). Unlike MLP experts, KAN experts directly learn trainable activation functions for each neuron.

## 4 EXPERIMENT

In this section, we validate the Bandit-MoE on three datasets and explore the impact of the Bandit Routing (BR) strategy and different expert networks on continual learning. All experimental results are obtained by averaging the results from three runs.

### 4.1 EXPERIMENTAL SETTING

**Datasets.** We evaluate the efficacy of Bandit-MoE by conducting extensive experiments on three benchmark datasets: CIFAR-100 Douillard et al. (2022), ImageNet-100, and TinyImageNet Yan et al. (2021). These datasets encompass 100, 100, and 200 distinct categories, respectively. The ImageNet-100 dataset is a subset of the ImageNet ILSVRC 2012 dataset Deng et al. (2009). Additional details about ImageNet-100 are provided in the Appendix A.2. We partition each dataset into distinct subsets, adopting a systematic approach referred to as "B$m$-$n$ steps", as shown in Tables 1, 2, 3, 4. The notation "B$m$" signifies that the experimental setup incorporates $m$ base categories as the initial task, while "$n$ steps" denotes the total number of incremental tasks introduced in the continual learning. As the value of $n$ increases, the sequence of tasks in the continual learning grows longer, thus intensifying the complexity of the learning process. For instance, in Table 2, the notation "B0-50 steps" specifies that the CIFAR-100 dataset begins with 0 base categories, and each incremental step introduces 2 new categories, culminating in a total of 50 sequential steps.

**Baselines.** We compare traditional continual learning methods, continual learning methods based on the CLIP model, and continual learning methods based on the MoE framework, as shown in Tables 1, 2, 3. In our comparative analysis, we evaluate several traditional continual learning methods,

Table 1: Comparison on the TinyImageNet dataset. The best results are indicated in bold and the second best results are underlined.

| Method | B100-20 steps | | B100-10 steps | | B100-5 steps | |
|---|---|---|---|---|---|---|
| | Last | Avg. | Last | Avg. | Last | Avg. |
| EWCKirkpatrick et al. (2017) | 4.73 | 12.35 | 3.79 | 15.82 | 6.00 | 19.01 |
| EEILCastro et al. (2018) | 29.72 | 40.41 | 34.64 | 45.03 | 35.12 | 47.17 |
| UCIRHou et al. (2019) | 30.85 | 42.84 | 37.29 | 48.58 | 39.42 | 50.30 |
| MUCLiu et al. (2020) | 10.32 | 21.89 | 15.33 | 26.67 | 19.20 | 32.23 |
| PASSZhu et al. (2021) | 32.93 | 42.01 | 39.27 | 47.19 | 41.64 | 49.54 |
| DyToxDouillard et al. (2022) | 36.21 | 46.18 | 42.79 | 52.26 | 47.23 | 55.58 |
| CLIP Zero-shot | 65.30 | 69.49 | 65.59 | 69.55 | 65.30 | 69.62 |
| CLIP Fine-tune | 44.55 | 54.62 | 41.54 | 57.05 | 46.66 | 61.54 |
| CLIP-LwF | 42.26 | 54.79 | 44.00 | 57.60 | 48.77 | 60.97 |
| CLIP-iCaRL | 64.68 | 69.65 | 67.05 | 74.12 | 70.89 | 77.02 |
| CLIP-LwF-VR | 63.89 | 69.94 | 67.05 | 74.12 | 70.89 | 77.56 |
| ZSCLZheng et al. (2023) | 68.30 | 77.18 | 71.62 | 78.61 | 73.57 | 80.27 |
| SEED Rypeść et al. (2024) | 41.44 | 51.46 | 42.67 | 49.97 | 43.86 | 48.95 |
| MoE-AdaptersYu et al. (2024) | 75.18 | 80.57 | 74.17 | 80.04 | 75.29 | 80.73 |
| Bandit-MoE | **77.25** | **82.02** | **76.61** | **81.48** | **77.67** | **81.88** |

Table 2: Comparison on the CIFAR-100 dataset.

| Method | B0-50 steps | | B0-20 steps | | B0-10 steps | |
|---|---|---|---|---|---|---|
| | Last | Avg. | Last | Avg. | Last | Avg. |
| UCIRHou et al. (2019) | 37.09 | 56.86 | 40.63 | 58.17 | 43.39 | 58.66 |
| BiCWu et al. (2019) | 41.04 | 62.09 | 47.02 | 66.48 | 53.54 | 68.80 |
| PODNetDouillard et al. (2020) | 32.99 | 51.19 | 35.02 | 53.97 | 41.05 | 58.03 |
| DERYan et al. (2021) | 59.76 | 72.05 | 62.55 | 73.98 | 64.35 | 74.64 |
| DyTox+Douillard et al. (2022) | 51.09 | 68.90 | 57.43 | 71.62 | 62.34 | 74.10 |
| DNEHu et al. (2023) | - | - | - | - | 70.04 | 74.86 |
| CLIP Zero-shot | 65.94 | 75.67 | 65.74 | 75.20 | 65.92 | 74.47 |
| CLIP Fine-tune | 18.89 | 39.23 | 43.13 | 59.69 | 53.23 | 65.46 |
| CLIP-LwF | 32.90 | 47.69 | 40.65 | 60.64 | 48.04 | 65.46 |
| CLIP-iCaRL | 59.07 | 71.28 | 64.55 | 73.32 | 70.97 | 79.35 |
| CLIP-LwF-VR | 59.07 | 71.02 | 63.54 | 74.54 | 70.75 | 78.81 |
| ZSCLZheng et al. (2023) | 67.36 | 79.92 | 69.58 | 80.39 | 73.65 | 82.15 |
| CLAPJha et al. (2024) | - | - | 70.01 | 70.65 | 61.35 | 67.88 |
| RAPFHuang et al. (2024) | - | - | - | - | 78.04 | 86.14 |
| MagMaxMarczak et al. (2024) | - | - | - | - | 79.00 | 85.63 |
| PROOFZhou et al. (2025) | - | - | 76.13 | 85.12 | 76.29 | 84.88 |
| SEED Rypeść et al. (2024) | 25.72 | 41.48 | 40.90 | 56.02 | 48.51 | 62.67 |
| MoE-Prompt Le et al. (2024) | - | - | 61.80 | 79.17 | 69.70 | 79.28 |
| MoE-AdaptersYu et al. (2024) | 72.32 | 82.92 | 76.30 | 84.69 | 78.22 | 86.38 |
| Bandit-MoE | **75.09** | **83.70** | **77.81** | **85.24** | **80.39** | **86.73** |

including UCIR Hou et al. (2019), BiC Wu et al. (2019), DER Yan et al. (2021), and EWC Kirkpatrick et al. (2017). As our proposed method is based on the CLIP model, we place particular emphasis on comparing it with CLIP-based continual learning methods. Specifically, we adapt and integrate three well-established continual learning techniques into the CLIP framework: LwF Li & Hoiem (2017), iCaRL Rebuffi et al. (2017), and LwF-VR Ding et al. (2022). These adapted methods are denoted as CLIP-LwF, CLIP-iCaRL, and CLIP-LwF-VR, respectively. The comparative analysis is designed to validate the efficacy of the proposed method itself, rather than attributing performance improvements solely to the advantages of CLIP's pretraining. In addition, we extend our comparative analysis to include other recent continual learning methods based on the MoE framework, such as SEED Rypeść et al. (2024) MoE-Prompt Le et al. (2024), and MoE-Adapters Yu et al. (2024). The details of SEED and MoE-Prompt are provided in the Appendix A.3.

**Metrics.** We evaluate the Bandit-MoE with two metrics: "Last" represents the average accuracy after the last class-incremental task; "Avg." Huang et al. (2024) represents the average accuracy across all class-incremental tasks. The formulations of the two metrics are provided in the Appendix A.4.

**Implementation Details.** In our experiments, we use the CLIP model with ViT-B/16 as the backbone. The experimental settings include a batch size of 64, 10 epochs, a total of $m = 5$ experts available for selection, with the top of $K = 2$ experts selected. This implies that expert selection is performed 128 times per experimental batch, thus the $N$ in Eq. 11 is 128. We employ the AdamW Loshchilov & Hutter (2019) optimizer with a learning rate of 1e-3. The MLP expert is configured as a single-layer fully connected architecture with a top layer output size of 768. For the LoRA expert, we employ a two-layer fully connected architecture

Table 3: Comparison on the ImageNet-100 dataset. The best results are indicated in bold and the second best results are underlined.

| Method | B0-20 steps | | B0-10 steps | |
|---|---|---|---|---|
| | Last | Avg. | Last | Avg. |
| UCIRHou et al. (2019) | - | - | 57.30 | 68.09 |
| TPCILTao et al. (2020) | - | - | 66.91 | 74.81 |
| DER(w/o P)Yan et al. (2021) | - | - | 66.70 | 77.18 |
| DERYan et al. (2021) | - | - | 66.06 | 76.12 |
| DyTox+Douillard et al. (2022) | - | - | 67.70 | 77.15 |
| CLIP-LwF | 60.52 | 76.57 | 69.6 | 81.63 |
| CLIP-iCaRL | 63.20 | 77.62 | 69.62 | 82.26 |
| CLIP-LwF-VR | 59.32 | 77.2 | 70.02 | 82.14 |
| SEED Rypeść et al. (2024) | 29.18 | 52.30 | 50.64 | 67.04 |
| MoE-AdaptersYu et al. (2024) | 72.15 | 84.53 | 73.92 | 84.71 |
| Bandit-MoE | **75.32** | **85.71** | **75.78** | **85.70** |

with an intermediate layer size of 64. The KAN expert consists of a two-layer KANlinear Li (2024) structure with an intermediate layer size of 32, a noise scale of 0.001, and a spline scale of 3. The MLP expert is configured as a single-layer fully connected architecture.

## 4.2 RESULTS AND COMPARISON

Across all the three datasets used in this study, the "Avg." and "Last" evaluation metrics outperform previous results, achieving optimal performance as shown in Tables 1, 2, 3. For example, the CIFAR-100 dataset, as shown in Table 1, the "Last" evaluation metric is approximately 2.7%, 1.5%, and 1.3% higher than the previous best results under the "B0-50 steps", "B0-20 steps", and "B0-10 steps" settings, respectively. Moreover, from the results, it can be observed that the performance improvement is generally more significant in long-sequence continual learning, demonstrating the potential of Bandit-MoE in handling complex continual tasks. More detailed experimental results and results for additional models are available in the Appendix A.3.

## 4.3 ABLATION STUDY

**Analysis of the BR algorithm.** The existing MoE routing strategy tends to preferentially select a few experts during training, leading to knowledge coverage and catastrophic forgetting, as shown in

Table 4: Validate the effectiveness of the BR.

| Method | CIFAR-100 B0-50 steps | | ImagetNet-100 B0-20steps | | TinyImagenet B100-20 steps | |
|---|---|---|---|---|---|---|
| | Last | Avg. | Last | Avg. | Last | Avg. |
| Baseline(w/ noise) | 73.48 | 82.94 | 73.26 | 83.87 | 74.73 | 80.45 |
| Baseline(w/ capacity) | 72.83 | 82.69 | 73.42 | 84.56 | 74.97 | 81.02 |
| Baseline(w/ BR) | **75.09** | **83.70** | **75.32** | **85.71** | **77.25** | **82.02** |

Table 5: Bandit-MoE with different expert structures on the "B0-10 steps" setting of the CIFAR-100 dataset.

| Method | 15-2 | | 10-2 | | 5-2 | | 2-2 | |
|---|---|---|---|---|---|---|---|---|
| | Last | Avg. | Last | Avg. | Last | Avg. | Last | Avg. |
| Bandit-MoE(w/ MLP) | 73.31 | 82.47 | 73.04 | 82.66 | 74.49 | 83.67 | 77.55 | 84.59 |
| Bandit-MoE(w/ KAN) | 75.99 | 84.29 | 77.13 | 84.79 | 76.98 | 85.34 | **78.87** | **86.02** |
| Bandit-MoE(w/ LoRA) | **77.49** | **85.13** | **77.99** | **85.37** | **80.39** | **86.73** | 77.95 | 86.05 |

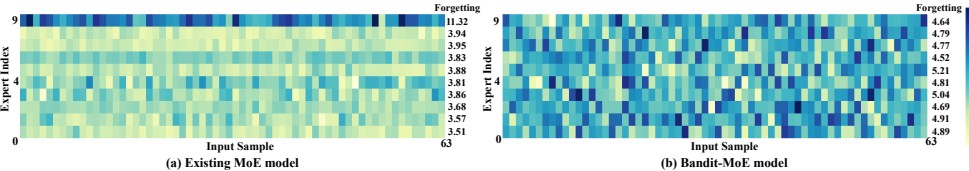

(a) Existing MoE model

(b) Bandit-MoE model

Figure 3: The expert selection probabilities of existing MoE models and the proposed Bandit-MoE model on the TinyImageNet dataset at the final layer, respectively. Darker colors indicate higher probabilities of expert selection. The values on the right show the forgetting rate of each expert. In (a), the existing model predominantly selects the ninth expert, resulting in knowledge overlap and a higher forgetting rate. This also leads to under-utilization of other experts. In contrast, (b) shows that Bandit-MoE leverages all experts more evenly to acquire diverse knowledge. This diverse knowledge acquisition is expected to enhance continual learning performance.

Figure 1 (a). Since long-sequence tasks are more challenging in mitigating catastrophic forgetting, we evaluate the effectiveness of the BR algorithm in the longest-sequence tasks across three datasets in Table 4. We compare models with different routing strategies. The "Baseline (w/ noise)" uses the routing strategy from the original MoE Shazeer et al. (2017). The "Baseline (w/ capacity)" uses the routing strategy of the Switch Transformers Fedus et al. (2022). It employs a capacity factor to distribute inputs evenly across all experts. As shown in Table 4, the BR algorithm outperforms the previous routing strategy in the "Last" metric on the CIFAR-100, ImageNet-100, and TinyImageNet datasets by 1.6%, 1.9%, and 2.3%, respectively. Furthermore, Figure 1 (b) shows that the BR algorithm can dynamically estimate the potential of experts, avoid prematurely discarding them, and select each expert fairly. These results demonstrate the effectiveness of the BR algorithm. Additional visual results are provided in the Appendix A.10.

**Analysis of expert network structures.** We evaluate three types of networks: MLP, LoRA, and KAN experts. The LoRA expert performs a low-rank decomposition of the weights in specific layers, enabling faster fine-tuning. Compared to the MLP and KAN experts, the LoRA expert has fewer parameters and lower computational cost. As shown in Table 5, "15-2" denotes a total of $m = 15$ experts with the top $K = 2$ selected. Except for the "2-2" setting, LoRA experts consistently outperform KAN experts, which, in turn, outperform MLP experts. In the "2-2" setting, however, KAN experts perform better than LoRA experts, with MLP experts consistently performing the worst. The suboptimal performance of the MLP expert can likely be attributed to its relatively simplistic architecture, which may limit its ability to model complex and non-linear relationships within the data. In contrast, LoRA and KAN experts exhibit different characteristics that potentially explain their respective behaviors. Specifically, we hypothesize that the KAN expert may possess superior representational capacity due to its architecture, which leverages kernelized attention to capture intricate patterns and dependencies in the data. In scenarios with fewer experts, the model can fully leverage KAN's capabilities, resulting in better performance. However, the KAN expert is more challenging to train and computationally expensive due to its larger number of parameters. Conversely, as the number of experts increases and tasks are subdivided into more class-incremental tasks or local features, LoRA experts outperform KAN experts. Further hyperparameter details for the KAN experts are available in Appendix A.5.

**Analysis of hyperparameter $\beta$.** We evaluate experiments on CIFAR-100 to find the optimal value of $\beta$ in Eq. 11 for all its settings, with each experiment running for 5 epochs. As shown in Table 6, the performance variation in different settings of hyperparameters is minimal, indicating good reproducibility in various CIFAR-100 setups. Based on these results, we recommend setting $\beta$ to 0.1. Furthermore, for long task sequences, reducing the number of training epochs often leads to better performance. This is because fewer classes per task increase the risk of overfitting when training for too long. For experiments on the hyperparameter $\beta$ for ImageNet-100 and TinyImageNet are provided in the Appendix A.5.

Table 6: Ablation experiments with parameter $\beta$ in the BR algorithm on the CIFAR-100 dataset.

| Method | 0.005 | | 0.1 | | 0.5 | | 1 | |
|---|---|---|---|---|---|---|---|---|
| | Last | Avg. | Last | Avg. | Last | Avg. | Last | Avg. |
| B0-10 steps | 79.35 | 86.68 | **80.39** | 86.73 | 80.05 | 86.62 | 79.29 | **86.78** |
| B0-20 steps | 76.30 | 85.15 | **77.58** | 85.21 | 77.42 | 85.24 | 77.37 | **85.28** |
| B0-50 steps | 73.16 | **83.63** | **74.26** | 83.40 | 73.58 | 83.50 | 73.76 | 83.34 |

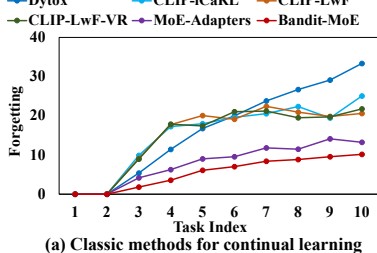

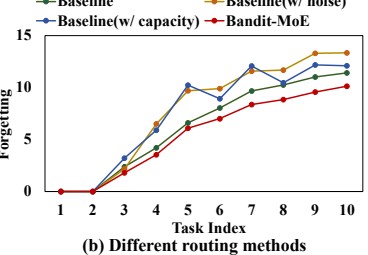

(a) Classic methods for continual learning     (b) Different routing methods

Figure 4: Figures (a) and (b) present experiments on "B0-10 steps" of the CIFAR-100 dataset. Figure (a) compares Bandit-MoE with other classic continual learning methods, and Figure (b) compares Bandit-MoE with other routing algorithms.

**Analysis of "Forgetting".** We utilize the "Forgetting" metric Chaudhry et al. (2018); Yu et al. (2024) to quantitatively assess the degradation in performance on previously learned tasks as new classes are incrementally introduced. Detailed calculations of this metric are provided in the Appendix A.4. Figure 3(a) demonstrates how knowledge coverage leads to catastrophic forgetting in current MoE models. Conversely, Figure 3(b) shows that our Bandit-MoE approach alleviates this issue by allowing each input to more equitably activate the most suitable expert. We show the forgetting resistance of the BR algorithm using this metric in Figure 4(a) and (b). Since the BR algorithm ensures fair expert selection and avoids overlapping knowledge among experts, it enables the model to retain diverse knowledge. Consequently, the resistance of Bandit-MoE to forgetting outperforms other methods across different class-incremental tasks. Additionally, we compare the Backward Transfer (BWT) and Forward Transfer (FWT) metrics Lopez-Paz & Ranzato (2017) among different models in the Appendix A.4.

**Analysis the effectiveness of Bandit-MoE in CL.** Bandit-MoE enhances continual learning performance in two key ways. First, it increases the selection probability of rarely-updated experts, thereby improving parameter utilization and balancing expert contributions, which effectively expands the model's capacity. To avoid the issue in MoE frameworks for continual learning where over-reliance on few experts, coupled with limited expert capacity, leads to severe forgetting. Second, the early stage of the training phase embeds knowledge with specific preferences into different experts. During continual learning, new samples are routed to the experts that perform best under the learned routing policy. This typically means they are routed to experts trained on similar data, which helps minimize parameter updates and mitigate forgetting while also encouraging expert diversity. More visualizations demonstrating how Bandit-MoE encourages fair expert selection and implicit expert diversity are provided in the Appendix A.6.

Furthermore, Ablation experiments on the number of experts hyperparameter are provided in the Appendix A.7. Bandit-MoE framework demonstrates competitive computational and spatial complexity. Comprehensive experimental results are available in Appendix A.8. To validate the versatility of our approach, we also conducted an evaluation of our framework in a non-continual learning context. The detailed experimental results are available in Appendix A.9.

## 5 CONCLUSIONS AND FUTURE

We propose Bandit-MoE framework, which pairs Bandit Routing (BR) with continual learning. The BR strategy estimates the upper confidence bound of each expert's expected gain, mitigating the risk of prematurely discarding experts that may underperform during the early stages of training due to insufficient optimization. This approach enables CL to acquire more diverse knowledge. A theoretical bound on estimation bias guarantees reliable routing. Using LoRA experts trims fine tuning cost and lessens forgetting. Experiments on three benchmarks show consistent gains over prior CL methods. Future work will explore adaptive top $K$ selection for greater flexibility.

## 6 ETHICS STATEMENT

This work adheres to the ICLR Code of Ethics. In this study, no human subjects or animal experimentation was involved. All datasets used, including CIFAR-100, ImageNet-100, and TinyImageNet datasets, were sourced in compliance with relevant usage guidelines, ensuring no violation of privacy. We have taken care to avoid any biases or discriminatory outcomes in our research process. No personally identifiable information was used, and no experiments were conducted that could raise privacy or security concerns. We are committed to maintaining transparency and integrity throughout the research process.

## 7 REPRODUCIBILITY STATEMENT

We have made every effort to ensure that the results presented in this paper are reproducible. The experimental setup, including training steps, model configurations, and hardware details, is described in detail in the paper. We have also provided a full description of Bandit-MoE, to assist others in reproducing our experiments. We believe these measures will enable other researchers to reproduce our work and further advance the field.

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

# A APPENDIX

## A.1 THE BANDIT ROUTING ALGORITHM

The pseudocode of the Bandit Routing algorithm is as follows:

---
**Algorithm 1** Bandit Routing

---
**Input:** $x, y, K$;
**Output:** Top$K$ experts
  1: Initialize $N, n_i$
  2: **for all** $i = 0, 1, 2, \cdots, Iter$ **do**
  3:    $r'_{i,j} \leftarrow \frac{\sum_{j=1}^{n_i} r_{i,j}}{n_i}$
  4:    $r_{i,j} \leftarrow r'_{i,j} + \beta\sqrt{\frac{2\ln N}{n_i}}$
  5:    Choose top$K$ experts
  6:    Calculate and update $N, n_i$
  7:    Calculate the output of model $\hat{y}$
  8:    $L = CE(y, \hat{y})$
  9:    Update $\boldsymbol{\theta}_R$
10: **end for**

---

## A.2 DETAIL OF IMAGENET-100

We choose 100 classes from ImageNet ILSVRC 2012 Deng et al. (2009) to build ImageNet-100 in the paper. The class names of ImageNet-100 are:

*'tench', 'goldfish', 'great white shark', 'tiger shark', 'hammerhead shark', 'electric ray', 'stingray', 'rooster', 'hen', 'ostrich', 'brambling', 'goldfinch', 'house finch', 'junco', 'indigo bunting', 'American robin', 'bulbul', 'jay', 'magpie', 'chickadee', 'American dipper', 'kite (bird of prey)', 'bald eagle', 'vulture', 'great grey owl', 'fire salamander', 'smooth newt', 'newt', 'spotted salamander', 'axolotl', 'American bullfrog', 'tree frog', 'tailed frog', 'loggerhead sea turtle', 'leatherback sea turtle', 'mud turtle', 'terrapin', 'box turtle', 'banded gecko', 'green iguana', 'Carolina anole', 'desert grassland whiptail lizard', 'agama', 'frilled-necked lizard', 'alligator lizard', 'Gila monster', 'European green lizard', 'chameleon', 'Komodo dragon', 'Nile crocodile', 'American alligator', 'riceratops', 'worm snake', 'ring-necked snake', 'eastern hog-nosed snake', 'smooth green snake', 'kingsnake', 'garter snake', 'water snake', 'ine snake', 'night snake', 'boa constrictor', 'African rock python', 'Indian cobra', 'green mamba', 'sea snake', 'Saharan horned viper', 'eastern diamondback rattlesnake', 'sidewinder rattlesnake', 'trilobite', 'harvestman', 'scorpion', 'yellow garden spider', 'barn spider', 'European garden spider', 'southern black widow', 'tarantula', 'wolf spider', 'tick', 'centipede', 'black grouse', 'ptarmigan', 'ruffed grouse', 'prairie grouse', 'peafowl', 'quail', 'partridge', 'african grey parrot', 'macaw', 'sulphur-crested cockatoo', 'lorikeet', 'coucal', 'bee eater', 'hornbill', 'hummingbird', 'jacamar', 'toucan', 'duck', 'red-breasted merganser', 'goose'.*

## A.3 BASELINES

SEED Rypeść et al. (2024) is a continual learning method based on the ResNet32 architecture, and all its experimental settings follow the original paper. MoE-Prompt Le et al. (2024) is a ViT-based continual learning method, which uses ViT-B/16 trained on the ImageNet 21K dataset as the backbone in its original paper. However, due to its strong pre-training ability and the presence of too many overlapping classes in the CIFAR-100 dataset, it is difficult to determine whether its resistance to forgetting from the model's continual learning ability or its pre-training capability. Therefore, we replace its backbone with ViT-B/16 trained on the ImageNet 1K dataset, while keeping other experimental settings consistent with the original paper.

In addition, we compare the experimental results of other baselines, as shown in Tables 7 and 8.

Table 7: Comparison on the CIFAR-100 dataset.

| Method | B0-20 steps | | B0-10 steps | |
|---|---|---|---|---|
| | Last | Avg. | Last | Avg. |
| SLCAZhang et al. (2023) | 66.84 | 78.96 | 67.58 | 80.53 |
| ADAM-AdapterZhou et al. (2023) | 58.12 | 70.18 | 65.50 | 75.76 |
| CLAP4CLIPJha et al. (2024) | 70.01 | 70.65 | 61.35 | 67.88 |
| Bandit-MoE | $77.81^{\pm 0.23}$ | $85.20^{\pm 0.05}$ | $79.53^{\pm 0.15}$ | $86.53^{\pm 0.10}$ |

Table 8: Comparison on the ImageNet-100 dataset.

| Method | B0-20 steps | | B0-10 steps | |
|---|---|---|---|---|
| | Last | Avg. | Last | Avg. |
| L2P++Wang et al. (2022b) | 62.10 | 75.43 | 67.22 | 80.51 |
| DualPromptWang et al. (2022a) | 61.10 | 75.40 | 67.38 | 80.65 |
| CODASmith et al. (2023) | 24.94 | 51.64 | 34.76 | 64.13 |
| SLCAZhang et al. (2023) | 63.36 | 78.40 | 59.92 | 78.63 |
| Bandit-MoE | $74.74^{\pm 0.24}$ | $85.01^{\pm 0.16}$ | $74.62^{\pm 0.37}$ | $85.02^{\pm 0.13}$ |

## A.4 EVALUATION METRICS

In continual learning tasks, BackWard Transfer (BWT) Lopez-Paz & Ranzato (2017) and ForWard Transfer (FWT) Lopez-Paz & Ranzato (2017) are important metrics for evaluating how well a model transfers knowledge between tasks.

**BWT.** BWT assesses the model's impact on its performance on previous tasks after learning a new task. A positive BWT indicates that the model's performance on previous tasks has improved during the learning process of the new task. Conversely, a negative BWT suggests that performance on previous tasks has deteriorated during this process, signifying the occurrence of forgetting.

$$\text{BWT} = \frac{1}{T-1} \sum_{i=1}^{T-1} \left( R_{T,i} - R_{i,i} \right), \tag{15}$$

where $T$ denotes the total number of class-incremental tasks, $R_{T,i}$ represents the model's accuracy on task $i$ after learning the final task $T$, and $R_{i,i}$ represents the accuracy on task $i$ immediately after learning that task (i.e., before subsequent tasks are learned). The larger the BWT value, the better. As shown in Table 9, Our experiments are conducted using the "B0-20 steps" setting on CIFAR-100. Our model achieves a BWT value of $-13.11$ and demonstrates superior resistance to forgetting.

**FWT.** FWT indicates whether the model exhibits positive knowledge transfer to subsequent tasks during the training of an earlier task.

$$\text{FWT} = \frac{1}{T-1} \sum_{i=2}^{T} \left( R_{i-1,i} - \bar{b}_i \right), \tag{16}$$

where $\bar{b}$ be the vector of test accuracies for each task at random initialization.

As shown in Table 9, Under the same experimental settings used for BWT, the FWT value of our model reaches $89.05$. The larger the FWT value, the better. This means that the model has better knowledge transfer ability.

**Forgetting.** The "Forgetting" metric measures the extent to which performance on previously learned class-incremental tasks deteriorates as new class-incremental tasks are learned. This metric is denoted by $f_{jk}$, and formulated by:

$$f_{jk} = \frac{1}{k-1} \sum_{j=0}^{k-2} \left( \max_{l \in [j,k-2]} a_{lj} - a_{kj} \right), \tag{17}$$

among them, $a_{lj}$ represents the accuracy of the model on the $j$-th task after learning the $l$-th task. $a_{kj}$ represents the accuracy of the model on the $j$-th task after learning the $k$-th task. $\max_{l \in \{1,...,k-1\}} a_{lj}$ represents the highest accuracy of the model on the $j$-th task across all past tasks. The $f_{jk}$ has a range of $[-1, 1]$, indicating the degree of forgetting of the model on the $j$-th task.

Table 9: Bandit-MoE with BWT and FWT on CIFAR-100 dataset.

| | EWC
Kirkpatrick et al. (2017) | DER++
Yan et al. (2021) | MoE-Adapters
Yu et al. (2024) | Bandit-MoE |
|---|---|---|---|---|
| BWT | -29.66 | -36.11 | -17.40 | **-13.11** |
| FWT | 43.29 | 39.62 | 89.00 | **89.05** |

## A.5 ABLATION EXPERIMENTS ABOUT PARAMETERS OF BANDIT-MoE

**Hyperparameter $\beta$.** We perform experiments on ImageNet-100, and TinyImageNet across all settings to explore the optimal value of $\beta$. Each experiment runs for 5 epochs. As shown in Tables 10, and 11, the performance across different hyperparameter settings shows minimal variation, indicating good reproducibility on ImageNet-100, and TinyImageNet. Based on the overall results, we recommend setting $\beta$ to 0.1. In addition, for long task sequences, reducing the number of training epochs tends to yield better performance, as fewer classes per task increase the risk of overfitting when training for too long.

Table 10: Ablation experiments with parameter $\beta$ in the BR algorithm on the TinyImageNet dataset.

| Method | 0.005 | | 0.1 | | 0.5 | | 1 | | 1.5 | | 2 | |
|---|---|---|---|---|---|---|---|---|---|---|---|---|
| | Last | Avg. | Last | Avg. | Last | Avg. | Last | Avg. | Last | Avg. | Last | Avg. |
| B100-5 steps | 77.32 | 81.61 | 77.56 | 81.89 | 76.96 | 81.56 | 77.34 | 81.64 | **77.67** | 81.88 | 77.33 | **82.19** |
| B100-10 steps | 76.45 | 81.25 | 76.61 | **81.27** | **76.66** | 81.22 | 76.4 | 81.29 | 76.56 | 81.08 | 76.36 | 81.22 |
| B100-20 steps | 75.85 | 80.72 | 75.64 | 80.63 | 75.72 | 80.38 | **76.44** | 80.65 | 75.37 | 80.71 | 76.25 | **80.81** |

Table 11: Ablation experiments with parameter $\beta$ in the BR algorithm on the ImageNet-100 dataset.

| Method | 0.005 | | 0.1 | | 0.5 | | 1 | |
|---|---|---|---|---|---|---|---|---|
| | Last | Avg. | Last | Avg. | Last | Avg. | Last | Avg. |
| B0-10 steps | 75.32 | **85.72** | 75.78 | 85.70 | 75.10 | 85.47 | **75.80** | 85.47 |
| B0-20 steps | 74.18 | 85.22 | **75.32** | 85.71 | 75.24 | **85.84** | 75.18 | 85.77 |

Table 12: Ablation experiments on parameter *scale spline* of KAN experts.

| Method | 1 | | 1.5 | | 2 | | 2.5 | | 3 | |
|---|---|---|---|---|---|---|---|---|---|---|
| | Last | Avg. | Last | Avg. | Last | Avg. | Last | Avg. | Last | Avg. |
| Bandit-MoE | 75.2 | 84.5 | 76.01 | 84.83 | 76.67 | 85.34 | 75.43 | 84.85 | **76.84** | **85.39** |

**Hyperparameter in KAN experts.** We explore two parameters of the KAN expert network: *scale spline* and *scale noise*. We conduct experiments on the "B0-10 steps" setting of CIFAR-100, as shown in Table 12. The experiments show that a *scale spline* parameter of 3 yields better results. The larger the scale spline parameter, the better the model's representation. As shown in Table 13, the experiments show that a *scale noise* parameter of 0.001 yields better results. The smaller the scale noise parameter, the more stable the model training becomes.

Table 13: Ablation experiments on parameter *scale noise* of KAN experts.

| Method | 0.001 | | 0.01 | | 0.05 | |
|---|---|---|---|---|---|---|
| | Last | Avg. | Last | Avg. | Last | Avg. |
| Bandit-MoE | **78.84** | **85.86** | 78.59 | 85.74 | 78.23 | 85.86 |

## A.6 ANALYSIS THE EFFECTIVENESS OF BANDIT-MoE

In continual learning, frequent parameter updates often lead to knowledge overwriting, which in turn causes catastrophic forgetting and degrades model performance. As illustrated in Figure 3(a), the

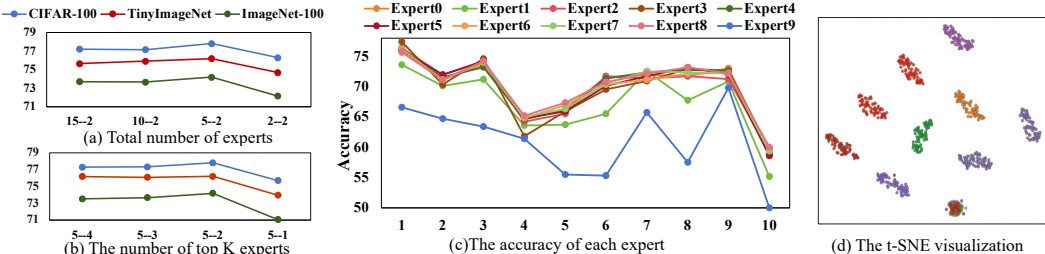

Figure 5: Figure (a) verifies the optimal total number of experts, while Figure (b) verifies the optimal number of top $K$ experts. The row axis represents different expert configurations, and the column axis represents model accuracy. Figure (c) presents the accuracy of each expert on 10 sequential tasks. Figure (d) shows the t-SNE visualization of expert inputs, based on an experiment conducted on CIFAR-100 with 5 experts in a "B0-10 steps" setting. Each color represents a single expert.

ninth expert is consistently selected, leading to severe knowledge overwriting. This results in its forgetting being significantly higher than that of other experts, which themselves remain underutilized. Simultaneously, Figure 5(c) shows that the ninth expert, due to knowledge overwriting, consistently exhibits the worst performance across every task.

To address this issue, we leverage the sparsity of MoE model to activate a distinct subset of experts for each input, thereby reducing interference. We further introduce a Bandit Routing strategy to estimate each expert's potential. This strategy enables a fairer selection of the most suitable experts for each input, which helps preserve knowledge. As shown in Figure 3(b), Bandit-MoE allows for a more equitable selection of each expert, preventing any single expert from experiencing severe catastrophic forgetting. Simultaneously, Figure 5(d) demonstrates that inputs received by different experts form distinct clusters. This indicates that similar inputs are processed by similar experts, implicitly promoting the diversity of expert knowledge.

Regarding the details of the BR strategy, it is embedded within the routing matrix $R$ and requires only one additional variable to track expert usage, without the need for an extra loss term. The number of experts remains constant regardless of the number of classes, thus avoiding additional computational costs.

## A.7 ABLATION EXPERIMENTS ABOUT EXPERT NUMBER

In Figure 5, we examine the impact of the total number of experts and the number of selected experts on the Bandit-MoE. We evaluate on "B0-20 steps", "B0-20 steps", and "B100-20 steps" of CIFAR-100, ImageNet-100, and TinyImageNet datasets, respectively, with $\beta$ set to 0.05. The results show that when the total number of experts is 5 and the number of selected experts is 2, the model performs best. Figure 5(a) investigates the effect of the total number of experts, where a configuration like "15–2" indicates 15 total experts with top-2 selected at each step. Figure 5(b) explores the impact of the number of selected experts. In Figure 5(a), the "2–2" setting performs the worst because it fails to leverage the sparsity advantage of MoE. This leads to severe knowledge interference in continual learning and results in the lowest final accuracy. Among "15–2", "10–2", and "5–2", performance differences are small, with "5–2" achieving the best result. This is because the model's expressiveness depends primarily on the number of activated experts rather than the total number. In Figure 5(b), the "5–1" setting performs the worst due to limited expressiveness from only one active expert. The "5–4", "5–3", and "5–2" settings perform similarly, but "5–2" performs best. Although "5–4" and "5–3" retain some sparsity, activating too many experts increases the risk of knowledge interference. Overall, "5–2" configuration achieves a favorable balance; its degree of sparsity is sufficient to protect expert specialization while the number of active experts retains adequate expressive capacity.

## A.8 COMPUTATIONAL AND SPATIAL COMPLEXITY

We verify the computational and spatial complexity of the Bandit-MoE model. We conduct different experimental settings on the CIFAR-100 dataset. It can be observed in Tables 14 and 15 that the

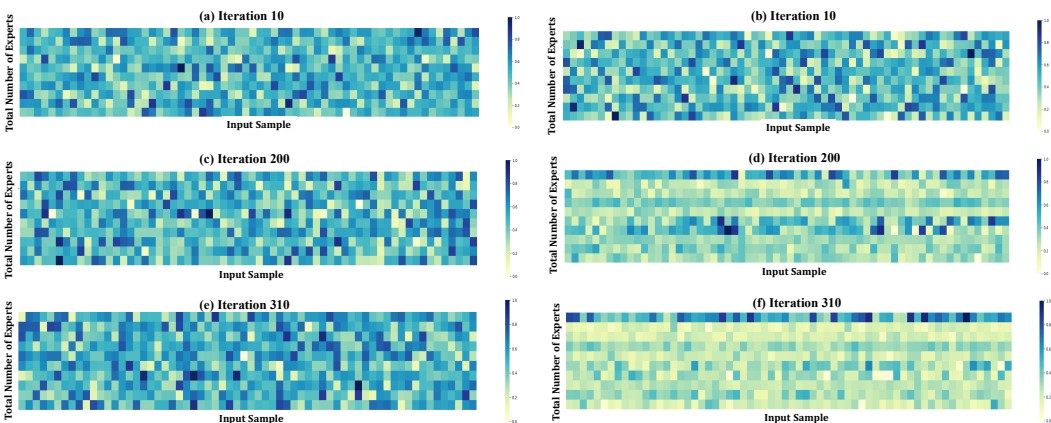

Figure 6: Heatmaps are generated for experiments on TinyImagenet at the "B100-10 steps" setting, with rows representing experts and columns representing input samples. These heatmaps visualize the last layer of the image router for the Bandit-MoE and MoE-Adapters models at different training periods. Figures (a), (c), and (e) show the heatmaps of the Bandit-MoE model at 10, 200, and 310 iterations, respectively. Similarly, Figures (b), (d), and (f) correspond to the MoE-Adapters model.

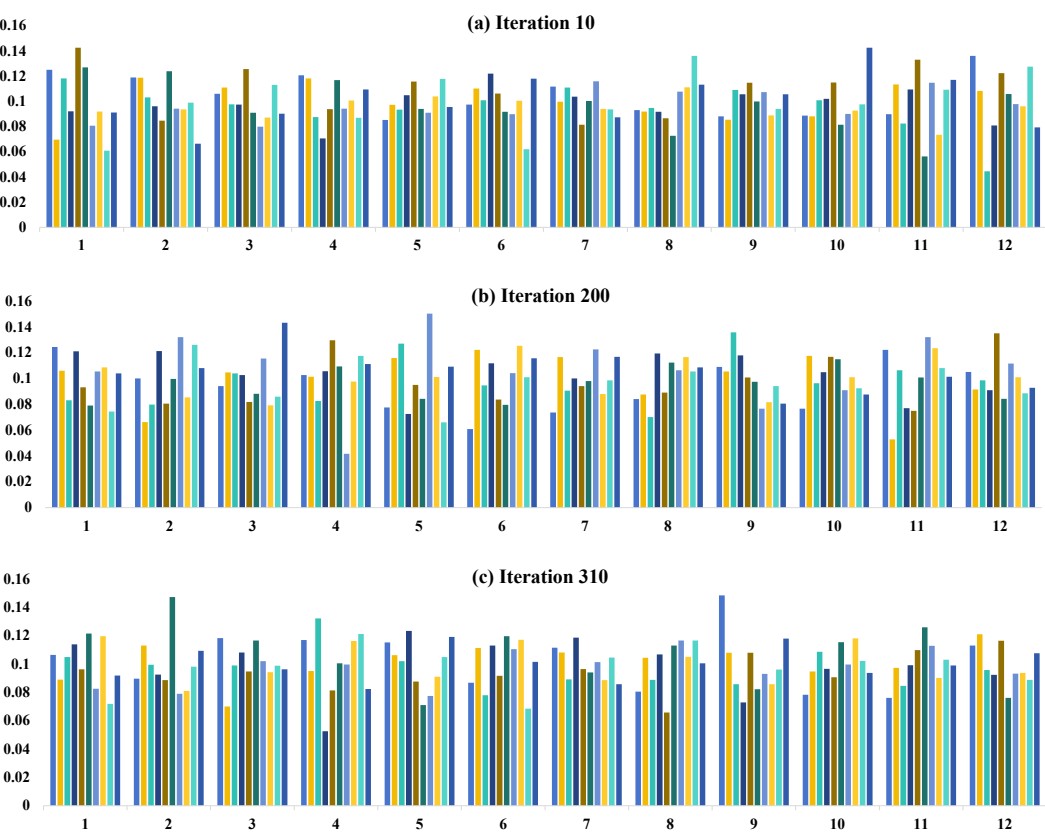

Figure 7: The histogram visualizes the probability of each expert being selected in 12 layers of the Bandit-MoE model on the TinyImageNet dataset at the "B100-10 steps" setting. Figures (a), (b), and (c) represent the visualizations for 10, 200, and 310 iterations, respectively. The rows of the histograms correspond to the number of model layers, while the columns represent the probability of each expert being selected. Each column in the histogram contains 10 segments representing 10 experts.

time and space occupied by our model are not significantly increased based on MoE-Adapters Yu et al. (2024).

Table 14: Memory usage of Bandit-MoE under different settings for the CIFAR-100 dataset.

| Method | B0-50 steps | B0-20 steps | B0-10 steps |
|---|---|---|---|
| MoE-Adapters Yu et al. (2024) | 11128MB | 11124MB | 11234MB |
| Bandit-MoE | 11204MB | 11210MB | 11334MB |

Table 15: The training and testing time of Bandit-MoE under different settings of CIFAR-100 dataset.

| Method | MoE-Adapters Yu et al. (2024) | | Bandit-MoE | |
|---|---|---|---|---|
| | Training | Testing | Training | Testing |
| B0-50 steps | 64.60s | 1.26s | 110.33s | 1.28s |
| B0-20 steps | 227.48s | 3.00s | 146.64s | 2.87s |
| B0-10 steps | 242.26s | 3.11s | 464.32s | 7.31s |

For the CIFAR-100 dataset in the "B0-50 steps" setting, with a batch size of 64 and 1 epoch, the Bandit-MoE model with 5 experts and top 2 selection requires 11204 MB of memory during inference and takes $24'11''$ for all inference class-incremental tasks. In comparison, the MoE-Adapters Yu et al. (2024) with 2 experts and top 2 selection uses 11128 MB of memory and takes $20'16''$ for all inference class-incremental tasks. Inference complexity slightly increases due to Bandit-MoE introducing the expected variance of rewards (the second term in Eq. 11, which requires computing this variance for 5 experts during each inference. The experimental results demonstrate that Bandit-MoE achieves superior performance while maintaining a computational complexity comparable to existing methods.

For model size complexity, we integrate MoE into each CLIP layer, with each expert adding 0.4 MB parameters and the routing strategy 0.3 KB. Compared to MoE-Adapters Yu et al. (2024), our approach introduces 14.4 MB (12 layers × 3 experts × 0.4 MB). Versus the CLIP model, it adds 24 MB (12 layers × (5 experts × 0.4 MB + 0.3 KB routing)).

A.9 NON-CONTINUAL LEARNING SETTINGS

We evaluate the generalizability of the Bandit-MoE framework in a non-continual learning setting. In Table 16, all models are based on ViT-B/16 backbone. The Bandit-MoE achieves optimal performance on all datasets. Preliminary experimental results indicate that Bandit-MoE is also applicable to non-continual learning, thereby demonstrating its versatility and generalizability.

Table 16: Bandit-MoE in non-continual learning settings across different datasets.

| Method | CIFAR-100 | TinyImagenet |
|---|---|---|
| CLIP Zero-shot | 77.9 | - |
| Swin Transformer Liu et al. (2021) | 82.02 | 69.09 |
| OneEncoder Faye et al. (2024) | 80.21 | 69.15 |
| MoE-Adapters Yu et al. (2024) | 88.60 | 83.79 |
| Bandit-MoE | **89.31** | **84.50** |

A.10 VISUALIZATION OF THE ROUTER

As shown in Figure 6, we visualize the last layer in image routers for the Bandit-MoE and MoE-Adapters models, respectively. The MoE-Adapters model gradually prefer one single expert due to the routing bias. In contrast, Bandit-MoE enables a fair selection of experts through the BR algorithm, allowing the model to acquire a more diverse set of expert knowledge to mitigate catastrophic forgetting.

Finally, we visualize the visual router of the Bandit-MoE model across iterations, showing the experts selecte at each layer, as depicted in Figure 7. Each layer has different experts with high importance, each addressing distinct aspects of the features.

## A.11 LARGE LANGUAGE MODEL USAGE STATEMENT

In the writing and editing of this paper, a large language model (LLM) was used as an assistive tool to improve clarity, refine phrasing, and enhance grammar. This usage was strictly limited to polishing the manuscript, and the LLM did not contribute to the scientific content, methodology, or core ideas of the research. All data, analysis, and conclusions presented in this work are entirely the result of the authors' own efforts.

