# OpenReview forum: "Bandit-MoE: Diverse Knowledge Acquisition through Bandit Routing for Continual Learning"
_ICLR.cc/2026/Conference — ICLR 2026 Conference Withdrawn Submission_

### Official Review · Reviewer_efLc · 2025-10-30

**Soundness:** 2
**Presentation:** 3
**Contribution:** 2
**Rating:** 2
**Confidence:** 3

**Summary:**

This paper addresses catastrophic forgetting in continual learning by proposing Bandit-MoE, a Mixture-of-Experts framework with Bandit Routing (BR) strategy. The key idea is to formulate expert selection as a Multi-Armed Bandit problem and use Upper Confidence Bound (UCB) algorithms to balance exploration and exploitation when routing samples to experts.

Main contributions:
1. Bandit Routing Strategy: Formulates MoE expert selection as MAB problem
2. Theoretical Guarantee: Proves using Hoeffding's Lemma that the UCB estimation bias is bounded with probability
3. Empirical Results: Achieves state-of-the-art performance on CIFAR-100, ImageNet-100, and TinyImageNet

**Strengths:**

1. Authors raise an important problem that existing MoE in CL routing concentrates on certain experts, leading to higher forgetting.
2. The paper is well-structured with visualizations (especially Figure 1 contrasting traditional MoE vs. Bandit-MoE). The writing is clear and well-accessible.
3. Experiments cover CIFAR-100, ImageNet-100, and TinyImageNet.

**Weaknesses:**

Major concerns

1. The theoretical proof (Section 3.2, lines 226-253) assumes stationary reward distributions. However, as the task changes, CL involves changing rewards. This assumption is fundamental to standard UCB convergence guarantees.

2. The proof assumes independent experiments. However, in the CL scenario, as the task changes, the joint distribution of (input, output) changes across tasks.

3. Unclear UCB implementation. The algorithm did not mention the inference logic, e.g., whether the UCB statistics are fixed after the last task training. Additionally, it's unclear what the exact definition of reward is. Line 191 claims that the router output is the reward, while Algorithm 1 in A.1 uses the cross-entropy loss, and the theorem assumes reward from [0,1]. This inconsistency makes the actual UCB implementation unclear and potentially incorrect..

Minor concern:
1. The paper did not discuss the non-stationary bandit methods, such as discounting or sliding window.

**Questions:**

Q1: UCB implementation in inference.

What is the inference routing calculation? Are the UCB statistics updated during the inference stage?

Q2: Non-stationarity

Why is the UCB suitable for the CL scenario where distribution is shifting?

---

### Official Review · Reviewer_yAsN · 2025-10-31

**Soundness:** 2
**Presentation:** 1
**Contribution:** 1
**Rating:** 2
**Confidence:** 3

**Summary:**

The paper addresses class-incremental learning by leveraging the multi-armed bandit method for Mixture-of-Experts (MoE). The authors also investigate the impact of expert structures on continual learning. Experiments based on the pre-trained model CLIP demonstrate the effectiveness of the proposed method compared to the baselines.

**Strengths:**

- The motivation is clear, as the multi-armed bandit approach can be useful for MoE routing.

- The experiments show that the proposed method, fine-tuned on CLIP, outperforms the baselines.

**Weaknesses:**

- The paper lacks novelty. The authors simply apply the multi-armed bandit method to MoE routing, which is neither surprising nor particularly innovative. The known issues with MoE, such as biased selection of a few experts, are already well established.

- The investigation into the effectiveness of the expert structures lacks depth, as it merely reports which structure performs best empirically without providing much theoretical insight into why.

- The paper’s structure needs improvement. The spacing between paragraphs, sections, subsections, and equations is too small, making it difficult to read.

- On page 5, the proof is presented without a clear theorem statement.

- Several sentences are unclear. For example, on page 5, line 253: “Therefore, our algorithm is reasonable.” is ambiguous.

- The experiments are not entirely convincing. It is unclear why the authors use CLIP, as the experiments do not seem to require text input. It would make more sense to use a simpler ViT model pre-trained on ImageNet or trained from scratch.

- The experimental setup requires more clarification. It is not clear whether all baselines are also based on CLIP. It appears that many baselines, such as EWC and EEIL, are trained from scratch. If that is the case, comparing them with Bandit-MoE fine-tuned on CLIP would be unfair.

**Questions:**

- [1] raises a concern that fine-tuning strong pre-trained models could cause information leakage from pre-training classes to continual learning tasks, making the comparison unfair. The authors mention removing classes similar to continual learning tasks during the pre-training stage. The authors should compare Bandit-MoE against this method.

[1] Learnability and Algorithm for Continual Learning, ICML 2023

---

### Official Review · Reviewer_nF5n · 2025-10-31

**Soundness:** 4
**Presentation:** 4
**Contribution:** 3
**Rating:** 4
**Confidence:** 4

**Summary:**

The authors observe that when using MoE for continual learning, the routing strategy tends to route tasks to certain experts, resulting in the overwriting of existing knowledge. The proposed Bandit-MoE solves the exploration and exploitation problem by utilizing the experts evenly with UCB. The method provides steady gains on several datasets across different expert structures.

**Strengths:**

1. The authors recognize the problem that routing strategy does not fully utilize the experts, and solve it with a bandit algorithm with theoretical analyses, which is well-motivated.
2. The method steadily provides performance gain on several datasets.
3. It is great that the author also investigated different expert structures, showing that BR can work on different architectures.

**Weaknesses:**

1. It is good that the authors show that Bandit-MoE does leverage the experts more evenly, but the actual gains compared to MoE-Adapters and other baseline strategies are lackluster. I suggest that the authors should leverage the method on other tasks to show its potential.
2. The load balancing problem when routing tasks to experts is not new. The authors should cite papers that also discuss the phenomenon of routing strategies favoring experts that trained on prior samples, such as [1] and [2].

[1] Chen, Hung-Jen, et al. "Mitigating forgetting in online continual learning via instance-aware parameterization." Advances in Neural Information Processing Systems 33 (2020): 17466-17477.

[2] Zhou, Yanqi, et al. "Mixture-of-experts with expert choice routing." Advances in Neural Information Processing Systems 35 (2022): 7103-7114.

**Questions:**

Please clarify whether Figure 5(c) represents the standard MoE or Bandit-MoE. This distinction should be clearly indicated in the figures. I assume it is the standard MoE, since Expert9 is consistently selected. If that is the case, please also include the accuracy of each expert after employing the BR strategy.

As I mentioned in the weakness sections, even after utilizing the experts better, the performance gain seems minimal, which is the biggest weakness of the paper. Please explain and provide an analysis of why this is the case. Alternatively, the author could apply the methods to a different dataset or scenario that can show the potential of the method. Overall, I think it is a solid work, but the performance gain is underwhelming.

---

### Official Review · Reviewer_qHGs · 2025-11-01

**Soundness:** 3
**Presentation:** 1
**Contribution:** 2
**Rating:** 2
**Confidence:** 4

**Summary:**

This paper presents a novel perspective by framing Mixture-of-Experts (MoE) methods with parameter-efficient tuning (PET) as a multi-armed bandit problem. It incorporates the UCB bandit routing strategy in the continual learning (CL) scenario. Its key contributions include a theoretical bound on the estimation bias of expert gains, compatibility with diverse PET architectures, and experiments on class-incremental learning benchmarks. The results show consistent performance improvements over previous methods, especially in long-sequence tasks, with enhanced knowledge preservation and reduced forgetting. However, the study remains largely built upon well-established strategies and does not sufficiently explore the implications of the multi-armed bandit perspective for CL problems. A more detailed methodology with clear definitions is also lacking. Additionally, although multimodal models are employed, the experiments are limited to image classification tasks, and the overall network architecture remains relatively simple and lacks generalizability.

**Strengths:**

1. The integration of multi-armed bandit principles into MoE routing provides a fresh perspective on balancing expert exploration and exploitation, and also helps to reveal the core problem of CL.
2. The introduction of UCB in MoE routing can dynamically estimate expert potential using reward expectation and variance, and proves to be effective in the CL scenario.
3. The proposed method is compatible with various adapter architectures, including MLP, LoRA, and KAN, allowing scalability and transferability.

**Weaknesses:**

1. The “method” section devotes considerable space to preliminary introductions, with Section 3.3 primarily summarizing past approaches rather than describing the paper's novel method. Conversely, regarding the paper's core contribution, *formulate the expert selection problem in MoE as a Multi-Armed Bandit problem*, Section 3.2 remains focused on the effectiveness of the UCB algorithm itself, as previously proposed by other researchers, without providing a detailed description or exposition of its application within the CL scenario.
2. The model heavily relies on a fixed version of CLIP (e.g., CLIP-ViT-B/16) for multimodal feature extraction, which limits flexibility and generalizability across diverse backbone architectures. Expanding experiments to include larger or more varied vision/text encoders would better demonstrate the robustness of the proposed expert selection framework when integrated with different multimodal foundation models.
3. The baseline comparisons are limited and do not fully reflect recent state-of-the-art methods. For instance, relevant advances in pretrained-model-based continual learning, such as RAIL[1] and CoDyRA[2] for vision-language models, and Hide-PET[3] for conventional class-incremental learning, are not included, weakening the validation of the proposed method's competitiveness.
4. (Minor issue) This draft requires further refinement and elaboration. For example, the term *expert gain* is not clearly defined in the context but is used throughout the manuscript, which may make it difficult for CL researchers lacking a relevant background to understand.

[1] Advancing cross-domain discriminability in continual learning of vision-language models
[2] Adaptive rank, reduced forgetting: Knowledge retention in continual learning vision-language models with dynamic rank-selective lora
[3] HiDe-PET: Continual Learning via Hierarchical Decomposition of Parameter-Efficient Tuning

**Questions:**

See the weakness part. Highly appreciate the authors' efforts. If my concerns are addressed with more evidence or further refinement to the manuscript, I'd be happy to increase the rating.

---

### Note · Authors · 2025-11-14

I have read and agree with the venue's withdrawal policy on behalf of myself and my co-authors.